# Comparative Study of the Synthesis of a Red Ceramic Pigment Using Microwave Heat Treatment

Eva Miguel [1], Guillermo Paulo-Redondo [1], Juan Bautista Carda Castelló [2] and Isaac Nebot-Díaz [1,*]

[1] Ceramic Technology Department, Escola Superior de Ceràmica de L'Alcora (ESCAL-ISEACV), 12110 L'Alcora, Spain; eva.miguel@escal.es (E.M.); guillermo.paulo@escal.es (G.P.-R.)

[2] Organic and Inorganic Chemistry Department, University Jaume I, 12006 Castelló de la Plana, Spain; carda@uji.es

[*] Correspondence: isaac.nebot@escal.es; Tel.: +34-964-399-450

**Abstract:** In this study, a new red ceramic pigment has been developed within a perovskite structure, and microwave heat treatments have been applied. Those red ceramic pigments within the $YAlO_3$ system doped with chromium with the nominal composition $Y_{0.98}Al_{0.98}Cr_{0.04}O_3$ were synthesized by traditional routes and alternative methods like coprecipitation. Also, heat treatment has been studied comparing a traditional electric and microwave kiln. Different flux agents have been incorporated to improve the synthesis reaction. Prepared pigments have been characterized by X-ray diffraction (XRD) as having a predominant phase of perovskite structure, which is responsible for the red shade, and a minority garnet phase that causes more brown colorations. Studies by Ultraviolet-Visible spectroscopy gave rise to a series of absorption bands that indicate the presence of Cr(III) in the octahedral position corresponding to perovskite and Cr(IV) corresponding to garnet in both the octahedral and tetrahedral positions. The perovskite phase is favored with the use of flux mix, corroborating the UV-visible results and being more pronounced in traditional high temperature thermal treatments. The coprecipitation route has been studied to increase the reactivity of the particles given their nanometric size; however, this reactivity favors a greater appearance of undesirable garnet phases with both types of flux. Scanning Electron Microscopy (SEM) micrographs offer information obtained from the secondary electrons of predominantly cubic crystalline phases with sizes between 1 μm and 2 μm in pigments synthesized via the traditional method and sizes less than 1μm together with the glassy phase in pigments synthesized via coprecipitation. Microwave thermal treatments have been studied, obtaining pigments with a majority structure of perovskite and garnet at lower temperatures and relatively short synthesis times. The feasibility of use in porous single-fired ceramic glazes has been studied, whose chromatic coordinates have been collected using an Ultraviolet-Visible Spectrophotometer based on the CIEL*a*b* system.

**Keywords:** perovskite; ceramic pigment; red pigment; coprecipitation; microwave





## 1. Introduction

There is currently great interest in obtaining ceramic glazes with a stable red shade in production systems, given the increasing demands in the production of porcelain stoneware and the impossibility of shade stability in high temperature cycles, added to the toxicity linked to certain pigments such as cadmium sulfoselenide. An alternative consists of pigments with the general formula $YAlO_3$ with a perovskite structure that present high chemical and thermal resistance whose structure is described as a compact cubic packing of anions, originating 12-coordination holes occupied by the cation A (Y(III)) of greater surrounded by 12 oxygen ions and a quarter of the octahedral voids formed by the cations B (Al(III) and Cr(III)) surrounded by 6 oxygen ions [1–3]. The bulky cation A (Y(III)) is housed in the gap formed by eight neighboring octahedrons that delimit a cube, whose incorporation $Y^{3+}$ (ionic radius = 0.89 Å) causes a distortion of the cubic structure to the tetragonal one. The red shade is obtained by doping the structure

with chromium so that ions of Cr(III) (ionic radius = 0.54 Å) replace some Al(III) cations (ionic radius = 0.62 Å) [4] causing an increase of the crystalline field on Cr(III) in octahedral coordination [5–7]. This phenomenon causes electronic transitions at higher energies between d orbitals, producing a red shade in the absorption spectrum. The intensity of the crystalline field on the Cr(III) chromophore ion is in turn intensified with the introduction of the $Y^{3+}$ cation by decreasing the Cr–O distance [6,8–11]. Phases with a YAG garnet structure $(Y_3Al_5O_{12})$ can originate within the $YAlO_3$ system, offering red-brown colorations with 2% molar doped chromium, although they present complexity to increase the crystalline field and difficulty in controlling the oxidation states of chromium in thermal processes, causing green-brown colorations [10,11]. In this case, Al(III) is found occupying both the octahedral and tetrahedral holes, and Y(III) is occupying the dodecahedral holes, crystallizing in a cubic habit so that each octahedron is connected with 6 tetrahedrons and each tetrahedron with 4 octahedrons. In this structure, Cr(III) can occupy the octahedral positions and Cr(IV) the tetrahedral positions. Enhancement of the crystalline field is achieved by codoping the structure with smaller ions such as Ca(II) or Mg(II) so that Cr(IV) (ionic radius = 0.55 Å) [12] can be incorporated into the octahedral positions [2,11–16].

The pigment synthesis method commonly used in the ceramic industry is carried out via the traditional route. This is since it is a relatively economical system from a productive point of view since the raw materials used are composed almost entirely of oxides of the elements to be reacted, the chemical reactions are less complex compared to other synthesis methods, and its elaboration is faster, speeding up production times. One of the drawbacks that it presents is the high calcination temperatures when dealing with solid-solid reactions, which are directly linked to the sintering start temperatures of each of the oxides involved. To favor reactivity, flux agents are usually added that start the reaction at lower temperatures, facilitating energy savings and reducing production times. There are other alternative methods of synthesis using the "bottom up" system, a system that allows the synthesis of nanoparticles by means of the assembly of elemental structures by chemical processes until an agglomerate of particles of controlled stoichiometry of nanometric size is achieved. This type of synthesis promotes the reaction of the reagents in a homogeneous and controlled way, lowering the reaction temperatures and allowing the use of applications in techniques where the particle size is decisive, as is the case with inkjet technology, where this study will be applied [17]. One of the examples is the synthesis by coprecipitation, a method that consists of the in situ coprecipitation of the precursors in the form of hydroxides to form a colloidal gel, whose heat treatment allows an interaction between particles on a nanometric scale. Synthesis mechanisms called "fast chemistry" can be obtained by microwave-assisted heat treatment [3,18–21]. Thanks to this mechanism, reaction times at high temperatures are reduced by increasing the synthesis speed, caused by a conversion of energy when the electromagnetic field of microwaves interacts with matter. In solid-state synthesis, as is the case with ceramic pigments, the dielectric capacity of the material or its absorbing capacity to transform electromagnetic interaction into the form of heat is crucial [22–24]. Thus, oxides such as $Al_2O_3$, $ZrO_2$, $Si_3N_4$, or AlN do not adequately couple with conventional microwaves (2.45 GHz), requiring the addition of a radiation susceptor that allows self-heating and causes a considerable increase in temperature, as is the case with SiC, graphite, or activated carbon [13].

In the case of preparing ceramic pigments to develop inks for digital inkjet applications, there are very specific requirements, such as a particle size of less than 1 micron and ease of dispersion and non-agglomeration once the ink is prepared [17].

The main novelty of this work is the application of an alternative synthesis methodology to the traditional ceramic process for obtaining pigments. The energy necessary for the formation of the perovskite structure is also applied quickly and without long retention times through the application of microwave radiation. These two characteristics allow for the obtaining of a nanoparticulate and easily dispersible pigment that favors the formulation of ceramic inks for inkjet [18].

## 2. Materials and Methods

Ceramic pigments with a perovskite structure doped with chromium $Y_{1-x}Al_{1-y}Cr_{x+y}O_3$ with x = 0.02 and y = 0.02 values have been prepared, seeking to check synthesis conditions [1]. Previous studies show the obtaining of pigments whose stoichiometry originates in a red shade with a high presence of perovskite phase. In the present work, the influence of the synthesis route and the effect of the flux with temperature on the synthesis reaction to favor the perovskite phase responsible for the red shade have been studied. A solid-state (traditional or "ceramic") and coprecipitation synthesis route has been used, using different types of flux agents and heat treatment conditions.

The reagents used in the traditional preparation have been aluminum hydroxide $(Al(OH)_3$, 99% purity, MERCK, Darmstadt, Germany), yttrium oxide $(Y_2O_3$, 99.5% purity, SIGMA-ALDRICH, Darmstadt, Germany), and chromium oxide $(Cr_2O_3$, 99% purity, PANREAC, Barcelona, Spain). Also, calcium fluoride $(CaF_2$, 99.5% purity, PANREAC) has been added as a flux agent [12] in 5% by weight, and a flux mix composed of 3% NaF (99.5% purity, MERCK, Darmstadt, Germany), 2% $MgF_2$ (99.9% purity, MERCK, Darmstadt, Germany), and $Li_2CO_3$ (99.6% purity, SIGMA-ALDRICH, Darmstadt, Germany) introduced at a total of 6% in both synthesis methods (traditional and coprecipitation) thermally treated in conventional high temperature cycles. The influence of flux agents in the microwave synthesis reaction has also been studied, comparing the two types of flux agents in the traditional and coprecipitation synthesis routes as well as the influence of the absence of flux in the coprecipitation route by this thermal treatment method.

### 2.1. Synthesis by Traditional Route

The reagents used in the traditional preparation have been aluminum hydroxide $(Al(OH)_3$, 99% purity, MERCK, Darmstadt, Germany), yttrium oxide $(Y_2O_3$, 99.5% purity, SIGMA-ALDRICH, Darmstadt, Germany), and chromium oxide $(Cr_2O_3$, 99% purity, PANREAC, Barcelona, Spain). The synthesized pigments have been obtained by solid-solid reactions using the traditional method. The raw materials have been mixed into a stoichiometric mixture [1]. The reagent mixture has been homogenized in a planetary ball mill with an acetone dispersing medium, drying the resulting material. The homogenized powder has been added separately to the two types of flux for its study and correctly mixed for its subsequent thermal treatment.

### 2.2. Synthesis by Coprecipitation Route

The reagents used in coprecipitation synthesis have been aluminum nitrate $(Al(NO_3)_3 \cdot 9H_2O$ 99.5% purity, SIGMA-ALDRICH, Darmstadt, Germany), chromium nitrate $(Cr(NO_3)_3 \cdot 9H_2O$ 99.5% purity, SIGMA-ALDRICH, Darmstadt, Germany), and yttrium solution obtained by dissolving yttrium oxide $(Y_2O_3$, 99.5% purity, SIGMA ALDRICH, Darmstadt, Germany) in nitric acid $(HNO_3$, 65% PANREAC, Barcelona, Spain) by reflux at 80 °C for 2 h [5]. All solutions were prepared at a concentration of 0.5 M and used in a stochiometric ratio.

Coprecipitation was carried out in a basic ammonia medium at pH = 9 to avoid fractional precipitation, resulting in a mixed hydroxide of $Cr(OH)_3$, $Al(OH)_3$, and $Y(OH)_3$ precipitated together from the precursors in solution. The addition is carried out with the application of heat and with constant stirring until a colloidal gel is obtained. The gel obtained has been filtered, washed, dried, and refined for subsequent heat treatment, adding the two types of flux agents mentioned above for conventional heat treatment and no flux agents for microwave-assisted treatment.

### 2.3. Conventional Heat Treatment

The powders obtained by both synthesis routes have been dosed in mullite crucibles and subjected to calcination cycles in a Nanetti Mod. FCN furnace at temperatures comprised between 1200 °C and 1500 °C with a heating rate of 10 °C/min and 360 min of

dwell time at maximum temperature. The fired material has been refined and washed in an agate mortar.

*2.4. Microwave Assisted Heat Treatment*

The microwave-assisted heat treatment was carried out by mixing stoichiometrically the appropriate quantities of the starting precursors used in the traditional route with the two types of flux separately in order to compare it with the heat treatment offered by the traditional route. In turn, the powder obtained by coprecipitation was homogenized to study it in the absence of flux. Also, a domestic microwave oven (800 W) was used to heat an $Al_2O_3$ capsule painted inside with SiC (Figure 1). In this capsule, different samples were located, and microwaves with 800 W power were applied for 15 and 30 min, reaching a maximum temperature of 1017 °C and 1185 °C, respectively. The temperature reached was determined using process temperature control rings (PTCR) type LTH (temperature range 970 °C–1250 °C) from FERRO.

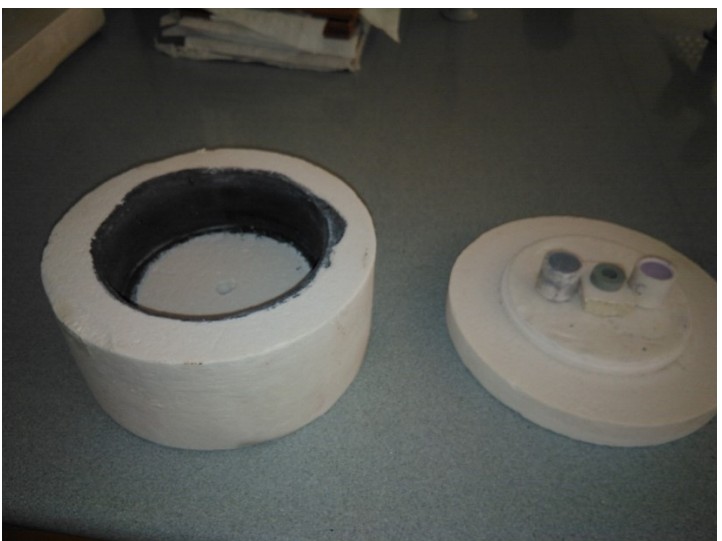

**Figure 1.** Capsule for a microwave kiln painted with SiC.

*2.5. Characterization of Samples*

The color development of all pigments prepared has been tested by introducing them at 4% by weight on a transparent industrial glaze and applying them to a single-fired ceramic test specimen with a glaze thickness of 0.4 mm. The glazed tiles have been subjected to an industrial cycle of 1100 °C with 5 min at maximum temperature and 50 min of total firing time (cold to cold).

The chromatic coordinates of the pigments obtained in this study and the glazed surfaces have been determined using an ultraviolet-visible spectrophotometer based on the CIE L*a*b* system (KONICA MINOLTA, CM-3600A, SpectraMagicNX d65 illumination, and a 2° observer). The crystalline phases present have been obtained by X-ray diffraction (BRUKER AXS, EndeavorD4, determination angles 5–70° 2θ with acquisition time 2 s. 0.05° 2θ). The absorption spectra at defined wavelengths have been determined by model ultraviolet-visible spectrophotometry using UV-Vis-NIR Jasco Mod. V670 spectrophotometer, measuring in the range of 200 nm and 800 nm.

**3. Results**

*3.1. Synthesis by Traditional Route*

Table 1 shows the chromatic coordinates of the glazed tiles obtained with the pigments synthesized by the traditional route with synthesis temperatures from 1300 °C to 1500 °C, showing shades from pink to reddish using $CaF_2$ flux and red colors using a flux mixture of NaF, $MgF_2$, and $Li_2CO_3$ at synthesis temperatures from 1200 °C to 1400 °C.

**Table 1.** Chromatic coordinate values (CIE-L*a*b* values) of glazed tiles with pigments synthesized by the ceramic route at different firing temperatures.

| Flux | T (°C) | Color | L* | a* | b* |
|---|---|---|---|---|---|
| | 1300 | Pink | 71.36 | 15.82 | 13.04 |
| $CaF_2$ | 1400 | Pink | 57.74 | 21.34 | 12.69 |
| | 1500 | Red | 60.50 | 36.06 | 23.24 |
| | 1200 | Red | 58.50 | 34.81 | 21.49 |
| $NaF + MgF_2 + Li_2CO_3$ | 1300 | Red | 54.82 | 23.68 | 11.41 |
| | 1400 | Red | 60.50 | 36.06 | 23.24 |

Figures 2 and 3 show the final appearance and tonality of the pigmented glazes obtained. The flux mix favors the synthesis reaction, obtaining interesting shades at a lower temperature than those obtained with $CaF_2$.

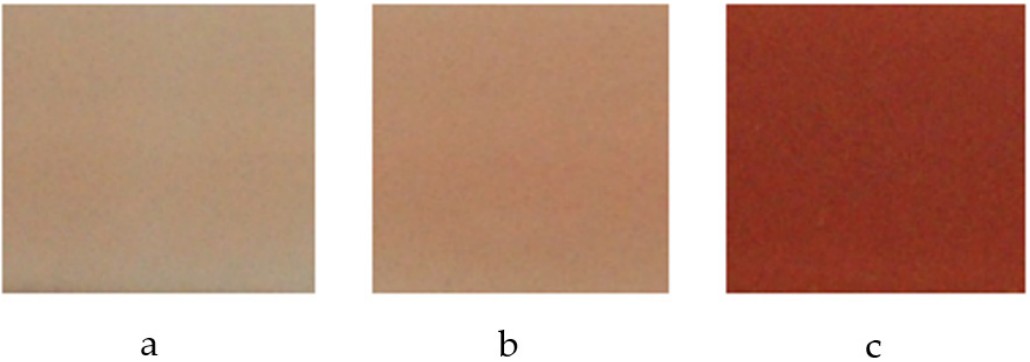

a       b       c

**Figure 2.** Color development of glazed tiles obtained by the traditional route with 5% *w/w* $CaF_2$ pigment $Y_{0.98}Al_{0.98}Cr_{0.04}O_3$ fired at different temperatures (**a**) 1300 °C; (**b**) 1400 °C; (**c**) 1500 °C.

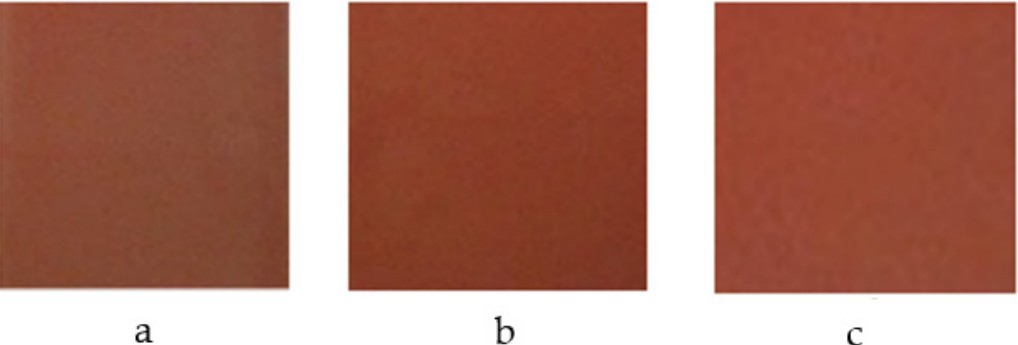

a       b       c

**Figure 3.** Color development of pigments obtained by traditional route with 6% *w/w* (NaF + $MgF_2$ + $Li_2CO_3$) pigment $Y_{0.98}Al_{0.98}Cr_{0.04}O_3$ fired at different temperatures (**a**) 1200 °C; (**b**) 1300 °C; (**c**) 1400 °C.

Figure 4 shows X-ray diffractions of the samples synthesized at 1400 °C with different flux agents. It can be determined that the predominant crystalline phase of perovskite $YAlO_3$ is responsible for the red coloration [1], together with minor phases of low-intensity garnet $Y_3Al_5O_{12}$. Studies show that with the increase in temperature, a higher intensity of the perovskite phase is obtained [12], and with it, a higher red shade, as shown in Table 1 and Figures 2 and 3. Both diffractograms present similar signals, with an increase in intensity of the perovskite diffraction peaks with the use of the flux mix.

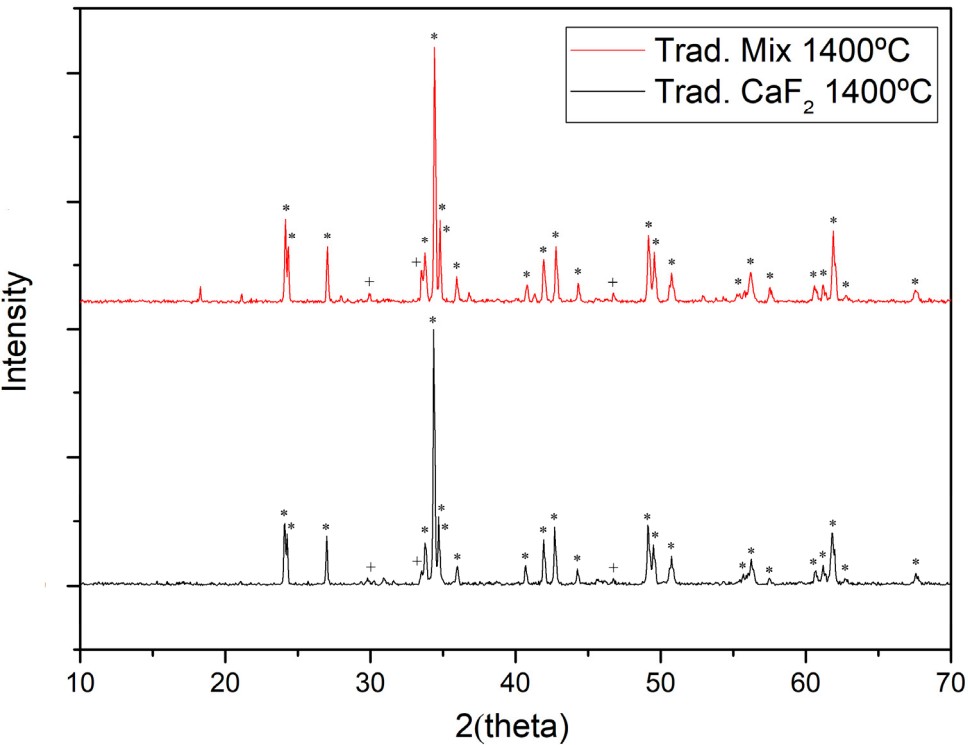

**Figure 4.** XRD of samples synthesized by the ceramic route and fired at 1400 °C with 5% *w/w* CaF$_2$ addition and 6% *w/w* (NaF + MgF$_2$ + Li$_2$CO$_3$) mix (+ garnet Y$_3$Al$_5$O$_{12}$ phase; * perovskite YAlO$_3$ phase).

Ultraviolet-visible spectrophotometry (Figure 5) exhibits the evolution of the absorption bands detected in the studied pigments, giving rise to broad bands in the range of wavelengths between 380 nm and 560 nm, within which is found the interval associated with the red shade of the pigment. visible spectrum. This band increases in intensity through the use of the flux mix, showing the formation of perovskite that can be identified by signals around 400 nm and 500 nm corresponding to Cr(III) in an octahedral environment with $^4A_{2g}(4F)\rightarrow^4T_{1g}(4F)$ electronic transitions and $^4A_{2g}(4F)\rightarrow^4T_{2g}(4F)$ reflecting in the red, as well as signals near 700 nm presenting spin-forbidden electronic transitions $^4A_{2g}(4F)\rightarrow^2T_{1g}(2G)$. A broad band between 400 nm and 550 nm is indicative of the presence of Cr(IV) in the garnet structure, superposed on Cr(III) signals, indicating the simultaneity of signals and coexistence of crystalline phases of perovskite and garnet. The formation of collateral garnet phases can in turn be identified by the presence of Cr(IV) in intervals between 400 nm and 550 nm, superimposed on Cr(III) signals, indicating the simultaneity of signals and coexistence of crystalline phases of garnet and perovskite. The presence of Cr(IV) in an octahedral environment can be associated with signals close to 480 nm with electronic transitions $^3T_{1g}(3F)\rightarrow^3T_{1g}(3P)$, to 380nm signals with $^3T_{1g}(3F)\rightarrow^3T_{1g}(3P)$ transitions, and at 530nm with transitions $^3T_{1g}(3F)\rightarrow^3T_{2g}(3F)$. The coexistence of Cr(IV) in tetrahedral coordination can be associated in the bands between 400 nm and 500 nm with $^3A_{2g}(3F)\rightarrow^3T_{1g}(3F)$ transitions, as well as being reflected in signals close to 480 nm with $^3A_{2g}(3F)\rightarrow^3T_{1g}(3P)$ transitions. Charge transfer bands between 250 nm and 270 nm are associated with Cr(IV) in octahedral coordination [14]. The transition $^3A_2\rightarrow^3T_2$ appears as a band in the visible range between 600 and 700 nm. The brown shade of systems with high concentrations of Cr(IV) in the garnet structure [1,2,9,10,12–14] is due to the latter and is more pronounced when using CaF$_2$.

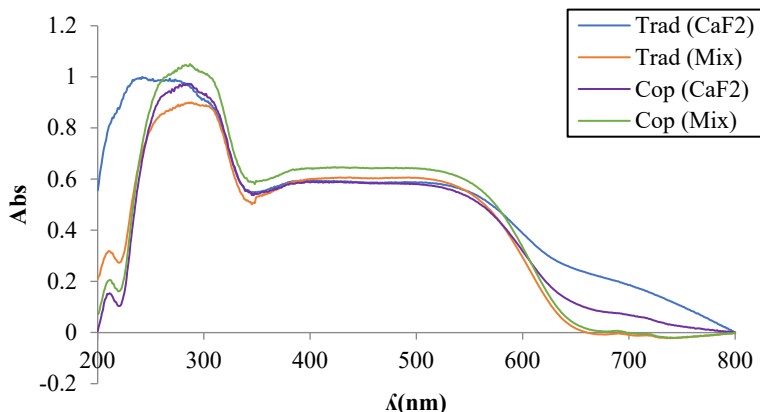

**Figure 5.** UV−Vis of samples synthesized by different synthesis routes fired at 1400 °C with 5% *w/w* CaF$_2$ addition (blue line corresponding to the traditional route and violet line corresponding to the coprecipiation route) and 6% *w/w* (NaF + MgF$_2$ + Li$_2$CO$_3$) mix (red line corresponding to the traditional route and green line corresponding to the coprecipitation route).

*3.2. Coprecipitation Synthesis*

Table 2 shows the chromatic coordinates of the pigments synthesized by the coprecipitation route. Compared with Table 1 (traditional route), it can be seen how the coordinate corresponding to the red shade (a*) is higher at lower firing temperatures when the flux mix is used.

**Table 2.** Chromatic coordinate values (CIE-L*a*b* values) of glazed tiles with pigments synthesized by the coprecipitation route at different firing temperatures.

| Flux | T (°C) | Color | L* | a* | b* |
|---|---|---|---|---|---|
| CaF$_2$ | 1400 | Red | 68.34 | 19.53 | 8.24 |
| | 1200 | Red | 63.00 | 27.62 | 18.97 |
| NaF + MgF$_2$ + Li$_2$CO$_3$ | 1300 | Red | 59.57 | 31.67 | 19.20 |
| | 1400 | Red | 60.98 | 28.33 | 15.00 |

Figure 6 shows the final appearance and tonality of the glazed tiles with pigments obtained by the coprecipitation route at 1400 °C. Flux mix favors the synthesis reaction, obtaining interesting shades at a lower temperature than those obtained with CaF$_2$.

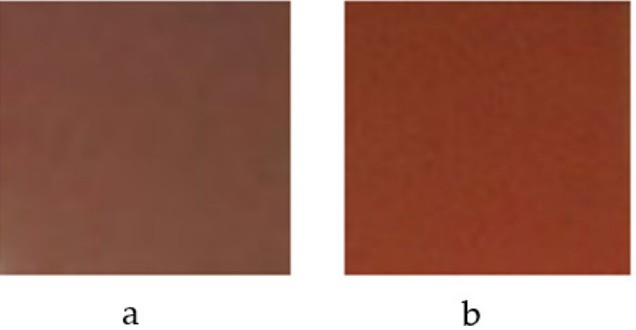

a        b

**Figure 6.** Color development of pigments obtained by coprecpitation synthesis with 5% *w/w* CaF$_2$ (**a**) and 6% *w/w* (NaF + MgF$_2$ + Li$_2$CO$_3$) pigment (**b**) Y$_{0.98}$Al$_{0.98}$Cr$_{0.04}$O$_3$ fired at 1400 °C.

The high reactivity of the nanoparticles in the synthesis by coprecipitation originates from perovskite phase signals of lower intensity compared to the traditional synthesis route and an increase in secondary garnet phases obtained by X-ray diffraction. The

crystallization process of the perovskite structure is favored in the synthesis using flux mix [5] (Figure 7), showing higher intensity signals.

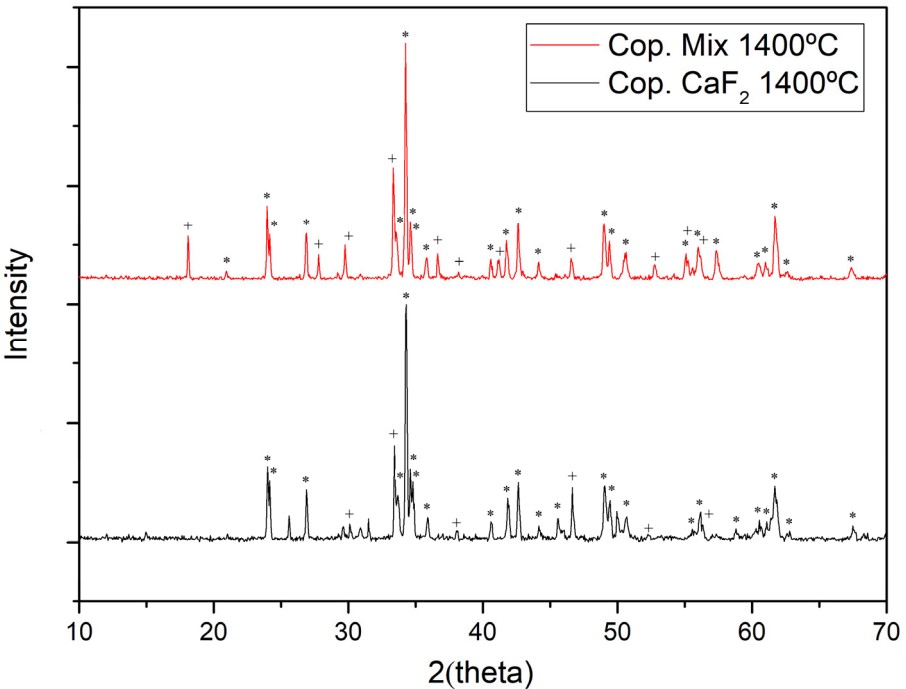

**Figure 7.** XRD of samples synthesized by coprecipitation synthesis fired at 1400 °C with 5% $w/w$ $CaF_2$ addition and 6% $w/w$ (NaF + MgF$_2$ + Li$_2$CO$_3$) mix (+ garnet $Y_3Al_5O_{12}$ phase; * perovsquite $YAlO_3$ phase).

Visible ultraviolet spectroscopy (Figure 5) shows bands similar to those obtained traditionally with both types of flux, originating a broad band between 400 nm and 500 nm corresponding to Cr(III) in an octahedral environment with electronic transitions $^4A_{2g}(4F) \rightarrow {}^4T_{1g}(4F)$ and $^4A_{2g}(4F) \rightarrow {}^4T_{2g}(4F)$ responsible for the red shade, of greater intensity with the flux mix compared to the traditional method. The charge transfer band between 250 nm and 270 nm associated with Cr(IV) in octahedral coordination increases with respect to the pigment obtained traditionally, presenting a broader band with the use of $CaF_2$. The $^3A_2 \rightarrow {}^3T_2$ transition associated with Cr(IV) in high concentrations in the garnet structure responsible for brown shade appears as a band in the visible between 600 and 700 nm of greater intensity with the use of $CaF_2$ compared to the traditional method.

Micrographs obtained by scanning electron microscopy show the morphology of the calcined pigment by the traditional route at 1400 °C, crystallized in a cubic shape (Figure 8). Crystal aggregates are observed, with crystallization ranging from slightly less than 1 μm to 2 μm. In the coprecipitation route, particles are highly sintered and partially glaze, and crystal size is less than 1 μm (Figure 9).

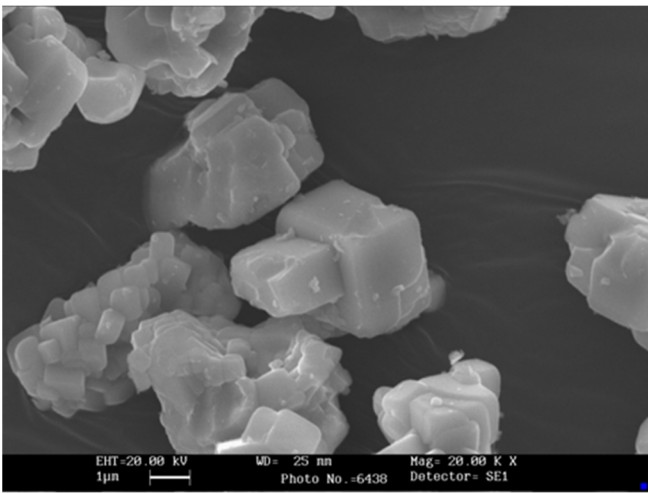

**Figure 8.** SEM micrography of pigment fired at 1400 °C synthesized by the ceramic route.

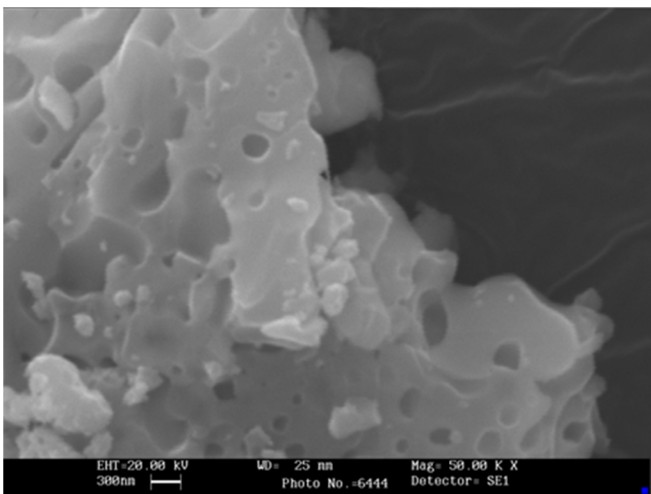

**Figure 9.** SEM micrography of pigment fired at 1400 °C synthetized by coprecipitation route.

*3.3. Microwave-Assisted Synthesis*

Table 3 shows the chromatic coordinates of the pigments synthesized by the traditional and coprecipitation routes using different flux agents and coprecipitation routes within flux, assisted by microwave heating at 1017 °C, and Figure 10 shows the final appearance of the glazed tiles. It has a light green coloration due to the low reactivity that occurred. The sample with the greatest a* value (red shade) is the coprecipitation sample.

**Table 3.** Chromatic coordinate values (CIE-L*a*b* values) of glazed tiles with pigments synthesized by traditional and coprecipitation routes assisted by microwave heating at 1017 °C.

| Synthesis Route | Flux | T (°C) | Color | L* | a* | b* |
|---|---|---|---|---|---|---|
| Ceramic | CaF$_2$ | 1017 | Light green | 78.96 | 0.31 | 7.13 |
| | NaF + MgF$_2$ + Li$_2$CO$_3$ | 1017 | Light green | 75.85 | −1.96 | 13.00 |
| Coprecipitation | NaF + MgF$_2$ + Li$_2$CO$_3$ | 1017 | Light Pink | 71.97 | 2.83 | 13.05 |

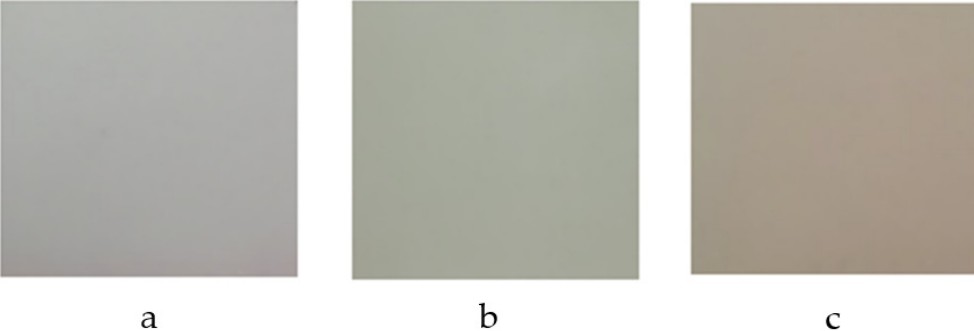

**Figure 10.** Color development of pigments obtained by 1017 °C traditional route synthesis with 5% *w/w* CaF$_2$ (**a**), 6% *w/w* (NaF + MgF$_2$ + Li$_2$CO$_3$) (**b**), and by coprecipitation synthesis within (**c**) pigment Y$_{0.98}$Al$_{0.98}$Cr$_{0.04}$O$_3$.

XRD (Figures 11 and 12) show the influence of the synthesis route and choice of the type of flux [5] on the reaction kinetics. The nanometric size of the coprecipitate favors obtaining the perovskite phase at low temperatures of 1017 °C in the presence of minor garnet phases (Figure 11). When firing temperatures increase up to 1185 °C, the intensity of the peaks associated with the perovskite increases (Figure 12). Through traditional synthesis, with the flux mix, crystalline phases of perovskite begin to appear at temperatures of 1017 °C, showing a diversity of unreacted Y$_2$O$_3$ signals of high intensity. However, at temperatures of 1185 °C, simultaneous signals appear with a predominant perovskite phase and the presence of a minority garnet phase, with diffraction signals like those obtained by coprecipitation. By using CaF$_2$ synthesized traditionally, at 1017 °C, most of the crystalline phase present is garnet, with the presence of perovskite and part of Y$_2$O$_3$. As the temperature increases to 1185 °C, the formation of perovskite increases slightly, although garnet phases predominate.

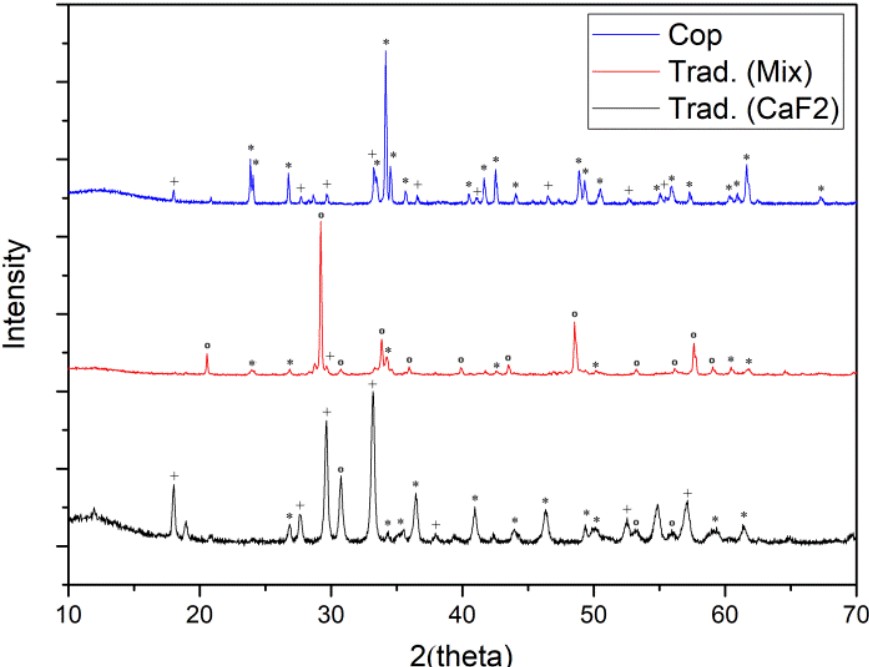

**Figure 11.** DRX of samples synthesized in microwave at 1017 °C by ceramic route with 5% *w/w* CaF$_2$ addition and 6% *w/w* mix flux NaF, MgF$_2$, and Li$_2$CO$_3$ mix addition and synthesized by coprecipitation route within flux (+ garnet Y$_3$Al$_5$O$_{12}$ phase; * perovskite YAlO$_3$ phase; ° Y$_2$O$_3$ phase).

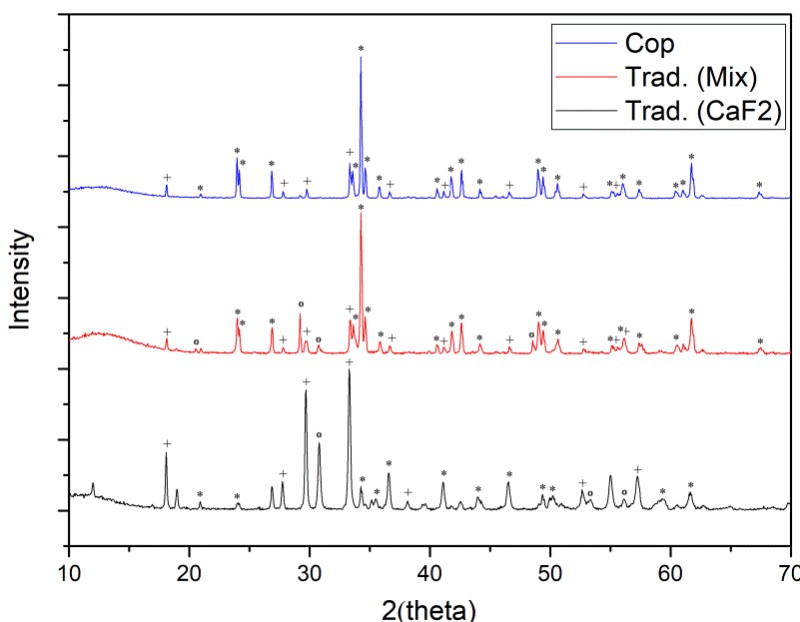

**Figure 12.** DRX of samples synthesized in microwave at 1185 °C by ceramic route with 5% *w/w* CaF$_2$ addition and 6% *w/w* mix flux NaF, MgF$_2$, and Li$_2$CO$_3$ mix addition and synthesized by coprecipitation route within flux (+ garnet Y$_3$Al$_5$O$_{12}$ phase; * perovskite YAlO$_3$ phase; ° Y$_2$O$_3$ phase).

Ultraviolet-visible spectrophotometry (Figure 13) of the pigments synthesized traditionally by microwave at temperatures of 1017 °C shows broad absorption bands at wavelengths between 380 nm and 560 nm, corresponding to Cr(III) in an octahedral environment with electronic transitions $^4A_{2g}(4F)\rightarrow^4T_{1g}(4F)$ and $^4A_{2g}(4F)\rightarrow^4T_{2g}(4F)$ with signals identical to those calcined in an electric furnace at 1400 °C of lower intensity, evidencing the need for a higher synthesis temperature. Said absorption band presents greater intensity with the flux mix, evidencing the greater formation of the perovskite phase compared to the garnet phase developed with CaF$_2$. In the coprecipitation pathway, the broad band is of low intensity, highlighting a signal close to 380 nm with transitions $^3T_{1g}(3F)\rightarrow^3T_{1g}(3P)$ corresponding to Cr(IV) in octahedral coordination. The $^3A_2\rightarrow^3T_2$ transition associated with Cr(IV) in the garnet structure in the range between 600 and 700 nm is greater with the use of CaF$_2$ via the traditional pathway. In the coprecipitation pathway, said signal is more accentuated in signals around 610 nm. The intensity of signals associated with Cr(IV) by charge transfer in the 270 nm and 300 nm intervals in all the samples synthesized by both routes stands out.

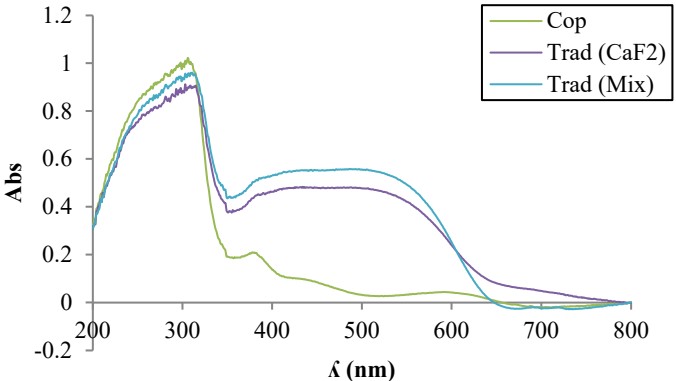

**Figure 13.** UV−Vis spectrophotometry of pigments synthesized in microwave at 1017 °C by ceramic route with 5% *w/w* CaF$_2$ addition and 6% *w/w* mix flux NaF, MgF$_2$, Li$_2$CO$_3$ mix addition and synthesized by coprecipitation route within flux.

### 3.4. Comparative Results with Other Studies

Table 4 provides information on X-ray diffraction results obtained in the study compared with data from the literature.

**Table 4.** Crystalline phases identified by X-ray diffraction at different synthesis conditions and comparison with previous bibliographic studies. P (Perovskite phase), G (Garnet phase), and Y (Ytria phase).

| Stoichiometry | Synthesis Route | T (°C)/dwell (h) | Flux | Crystalline Phase [Ref] |
|---|---|---|---|---|
| $Y_{0.98}Al_{0.98}Cr_{0.04}O_3$ | Ceramic | 1400/6 | $CaF_2$ | P >>> G |
| | | | $NaF + MgF_2 + Li_2CO_3$ | P >>>> G |
| $Y_{0.98}Al_{0.98}Cr_{0.04}O_3$ | Coprecipitation | | $CaF_2$ | P >> G |
| | | | $NaF + MgF_2 + Li_2CO_3$ | P >>> G |
| $Y_{0.98}Al_{0.98}Cr_{0.04}O_3$ | Ceramic + microwave heating | 1017 | $CaF_2$ | G >>> P >> Y |
| | | | $NaF + MgF_2 + Li_2CO_3$ | Y >>> P |
| | Coprecipitation + microwave heating | | - | P >> G |
| | Ceramic + microwave heating | 1185 | $CaF_2$ | P >> G > Y |
| | | | $NaF + MgF_2 + Li_2CO_3$ | P >>> G |
| | Coprecipitation + microwave heating | | - | P >>> G |
| $Y_{0.99}AlCr_{0.01}O_3$ | Ceramic | 1500/6 | $CaF_2$ | P >> G [1] |
| $Y_{0.97}AlCr_{0.03}O_3$ | | | | P >>> G [1] |
| $Y_{0.95}AlCr_{0.05}O_3$ | | | | P >>> G [1] |
| $Y_{0.90}AlCr_{0.10}O_3$ | | | | P >> G [1] |
| $Y_{0.98}Al_{0.98}Cr_{0.04}O_3$ | Ceramic | 1300/6 | $CaF_2$ | P >> G [1] |
| | | 1400/6 | | P >>> G [1] |
| | | 1500/6 | | P >>> G [1] |
| $Y_{0.97}AlCr_{0.03}O_3$ | Ceramic | 1300/6 | $CaF_2$ | P >> G [12] |
| | | 1400/6 | | P >>> G [12] |
| | | 1500/6 | | P [12] |
| $YAl_{0.97}Cr_{0.03}O_3$ | Ceramic | 1300/6 | $CaF_2$ | P >> G [12] |
| | | 1400/6 | | P >>> G [12] |
| | | 1500/6 | | P >>> G [12] |
| $YAlO_3$ | Coprecipitation | 1400/4 | - | P > G [5] |
| | | | $NaF + MgF_2 + Li_2CO_3$ | P [5] |
| $YAl_{0.97}Cr_{0.03}O_3$ | Coprecipitation | 1400/4 | $NaF + MgF_2 + Li_2CO_3$ | P [5] |

## 4. Conclusions

Ceramic pigments of red coloration have been synthesized within the $YAlO_3$ system doped with chromium with the nominal composition $Y_{0.98}Al_{0.98}Cr_{0.04}O_3$ through the traditional synthesis route, or "ceramic route", and via coprecipitation using $CaF_2$ and a flux mix NaF(1% *w/w*), $MgF_2$(2% *w/w*), and $Li_2CO_3$(3% *w/w*) as mineralizers at synthesis temperatures from 1200 °C to 1500 °C in an electric furnace and at temperatures between 1017 °C and 1185 °C in microwave-assisted synthesis. The samples have been applied in an industrial porous single firing cycle, and their L*a*b* colorimetric coordinates have been read in an Ultraviolet-visible Spectrophotometer following the Cie-L*a*b* model.

In the X-ray diffractions of the tests carried out, two crystalline phases of perovskite $YAlO_3$ (responsible for the red coloration) and garnet $Y_3Al_5O_{12}$ appear, the latter being a minority phase. The colorimetric results of the pigments synthesized via the traditional route offer lower L*(luminosity) values, indicative of a greater color intensity, and higher a*(red) values compared to the coprecipitation route. After analyzing the X-ray diffraction, a perovskite phase of greater intensity predominates, while in coprecipitation the signal intensity decreases, appearing more garnet. This is due to the fact that as the particle size decreases, the reactivity of the solid–solid reactions increases, favoring the more thermodynamically stable phases of garnet but making them undesirable. These data observed by X-ray diffraction are corroborated with the Ultraviolet-Visible spectrophotometric results. Spectrophotometry could identify octahedral Cr(III) signals belonging to the perovskite structure that increase in intensity as occurs in X-ray diffraction, as well as Cr(IV) signals in tetrahedral position and Cr(IV) in octahedral position belonging to the garnet structure. The synthesis temperature reached by calcining in an electric oven and assisted by microwaves is decisive for obtaining the perovskite structure; high synthesis temperatures are necessary for its correct crystallization. The addition of flux agents is necessary in order to decrease the synthesis temperature by the traditional route and accelerate the solid-solid reaction adequately. The mixture of mineralizers composed of NaF (1% $w/w$), $MgF_2$ (2% $w/w$), and $Li_2CO_3$ (3% $w/w$) is the one that favors the formation of perovskite with a higher degree of purity at temperatures relatively low from 1185 °C to 1400 °C, where the perovskite structure presents abundant signals in X-ray diffraction and absorption bands of greater intensity at wavelengths between 380 nm and 560 nm corresponding to the red coloration of the visible spectrum. By using a $CaF_2$ mineralizer at temperatures of 1185 °C, the first crystalline structure identified in x-ray diffraction is garnet, which evolves with temperature until it becomes perovskite at studied temperatures of 1400 °C, considerably decreasing the garnet phase and becoming a minority phase, indicating its reaction power at high temperatures. In the absence of mineralizers, perovskite phases (identified in X-ray diffraction) begin to develop at low temperatures of 1017 °C in pigments synthesized via coprecipitation, indicating a high degree of reaction. Several studies show the need for flux agents at high temperatures to obtain the crystalline structures of pervoschite, since without flux, even with the coprecipitation route, the secondary garnet phase appears with greater intensity [4,5]. The present study reflects the importance of the choice of the type of flux to favor the perovskite phases responsible for the red shade with temperature and the synthesis reaction at low temperatures. It has been observed that flux mix favors the appearance of perovskite at low temperatures compared to $CaF_2$ in both traditional and coprecipitation synthesis methods. In the case of coprecipitation, if compared with similar synthesis methods, according to studies [5], the perovskite phase is disadvantaged with the appearance of garnet. However, at low temperatures (microwave), in the absence of flux, perovskite signal intensity increases as the reaction increases due to increased reactivity given the nanometer size of the powder. In scanning electron microscopy (SEM), particle sizes below 1 µm up to 2 µm are observed in the traditional synthesis route, with crystalline formations of cubic morphology that could indicate the presence of perovskite with cubic structure distorted to orthorhombic or the garnet phase that crystallizes in cubic form. When observing the samples synthesized via co-precipitation, the pigment particles appear very sintered and partially melted, with crystals of size less than 1 µm, so that at the calcination temperature, they lose color intensity after application and firing of the glaze.

It is highly necessary to continue with the study of the application of microwaves with a longer dwell time to favor the development of the phases responsible for the color.

**Author Contributions:** Conceptualization, E.M., J.B.C.C. and I.N.-D.; Methodology, J.B.C.C. and I.N.-D.; Investigation, E.M. and G.P.-R.; Writing original draft preparation, E.M.; Writing review and editing, I.N.-D.; Supervision, J.B.C.C. All authors have read and agreed to the published version of the manuscript.

**Funding:** This research received no external funding.

**Institutional Review Board Statement:** Not applicable.

**Informed Consent Statement:** Not applicable.

**Data Availability Statement:** Not applicable.

**Conflicts of Interest:** The authors declare no conflict of interest.

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
