# Peer review of "Comparative Study of the Synthesis of a Red Ceramic Pigment Using Microwave Heat Treatment"

_2079-6447, doi:10.3390/colorants2030025_

Round 1

Reviewer 1 Report

In this work, the authors investigate the Comparative study of the synthesis of a red ceramic pigment using microwave heat treatment. The work is very interesting and with a high impact on literature. The manuscript can be accepted for the publication in Colorants after addressing the following major comments.

Below I present the observations related to the content.

1. The novelty of this study should be inserted in the text clearly.

2. The abstract need to rewrite and the author/s should focus on the main findings and the main results.

3. The lengthy sentences may be split in to smaller sentence without change of its meaning.

4. Also, suggested to include the  references in the introduction part. In the introduction can be completed with properties and application of nano-size ceramics. The discussion can be extended by using the following paper: DOI: 10.1016/j.jallcom.2020.156695

5. At the end of the introduction, the authors described the purpose (objectives) of the study, mentioning also the methods used, obviously briefly, but this should be explained in more detail in the methods.

6. The methodology section is not well organized for the readers to understand the concept.

7. The author/s should be try to compare the results of the current study with the results of other researchers and it is required to explain if there is any agreement or disagreement.The results obtained should be compared with the in the literature. More care should be taken to present the results and facilitate understanding of the work

8. Figure 1 (XRD)  has very poor resolution and is very loaded, to simplify and improve the resolution.

9. Compare XRD results with other articles.

10. The XRD parameters  compared with other literature studies.  Present the results obtained but do not compare with other existing studies. Are the values higher or lower than the literature? Add a comparative table.

11. Why is the size of the particles larger than the size of the crystallites?

12. In the conclusions appear aspects that should appear in the chapter of results and be described only strictly the conclusions of this study. In case of conclusions, you could explain what does it add to the subject area compared to other published material?

13. References are not written according to the guide (not all authors are listed in all references, they are not marked with initials, the titles of the articles and journals are not mentioned in many places, the volumes or pages are missing).

14 The language of the whole manuscript is not very good and an improvement on the English language needs to be done in order to improve language of the manuscript.

The language of the whole manuscript is not very good and an improvement on the English language needs to be done in order to improve language of the manuscript.

Author Response

Thank you very much for your review and for your effort doing it.

I would like to answer your requirements

We have rewrite the article with the help of a native english . You can find abstract has been rewritten with main results obtained.

Some new references have been added and methodology section has been reorganized. Also, results with XRD and UV-Vis spectrophtometry have been compared with other references

Thank you in advance for your help

Reviewer 2 Report

Miguel et al presented a comparative study of the synthesis of a red ceramic pigment using microwave heat treatment. Red color pigments within the YAlO3 system doped with chromium with nominal composition Y0.98Al0.98Cr0.04O3 were synthesized by traditional route and alternative methods like co-precipitation, thermally treated in an electric oven and by microwave radiation have been studied. Prepared pigments have been characterized by X-ray diffraction, with a predominant phase of perovskite structure responsible for the red coloration, together with a minority garnet phase, and by ultraviolet-visible spectroscopy, giving rise to a series of absorption bands that indicate the presence of Cr(III) in octahedral position and Cr(IV) in both octahedral and tetrahedral position. The work is interesting. It will provide valuable addition to the field of novel ceramic pigments. I recommend publication of this work in colorants after some revisions:

1. Further editing of English language is required.

2. The quality of the Figures (e.g., Figures 7 and 11) should be improved in the manuscript.

3. The references are too old in the manuscript, and they should be updated.

4. Some recent works on the transition-metal-based pigments can be added for more balanced reference citation: (e.g., Adv. Optical Mater. 2020, 8, 2000985, etc.).

Minor editing of English language should be required.

Author Response

(The authors gave the same response as above.)

Round 2

Reviewer 1 Report

The authors did not respond punctually to the 14 observations made in the first revision. To the reply to the reviewers, they attached only the revision 1 manuscript. I ask you to respond punctually and to take into account the observations suggested in the first revision as well.

Below I present again the observations related to the content.

1. The novelty of this study should be inserted in the text clearly.

2. The abstract need to rewrite and the author/s should focus on the main findings and the main results.

3. The lengthy sentences may be split in to smaller sentence without change of its meaning.

4. Also, suggested to include the  references in the introduction part. In the introduction can be completed with properties and application of nano-size ceramics. The discussion can be extended by using the following paper: DOI: 10.1016/j.jallcom.2020.156695

5. At the end of the introduction, the authors described the purpose (objectives) of the study, mentioning also the methods used, obviously briefly, but this should be explained in more detail in the methods.

6. The methodology section is not well organized for the readers to understand the concept.

7. The author/s should be try to compare the results of the current study with the results of other researchers and it is required to explain if there is any agreement or disagreement.The results obtained should be compared with the in the literature. More care should be taken to present the results and facilitate understanding of the work

8. Figure 1 (XRD)  has very poor resolution and is very loaded, to simplify and improve the resolution.

9. Compare XRD results with other articles.

10. The XRD parameters  compared with other literature studies.  Present the results obtained but do not compare with other existing studies. Are the values higher or lower than the literature? Add a comparative table.

11. Why is the size of the particles larger than the size of the crystallites?

12. In the conclusions appear aspects that should appear in the chapter of results and be described only strictly the conclusions of this study. In case of conclusions, you could explain what does it add to the subject area compared to other published material?

13. References are not written according to the guide (not all authors are listed in all references, they are not marked with initials, the titles of the articles and journals are not mentioned in many places, the volumes or pages are missing).

14 The language of the whole manuscript is not very good and an improvement on the English language needs to be done in order to improve language of the manuscript.

The language of the whole manuscript is not very good and an improvement on the English language needs to be done in order to improve language of the manuscript.

Author Response

Thank you very much for your review and for your effort doing it. I want to apologize because in previous answering request I failed attaching you the correct answer.

I would like to answer you the questions you made us

  1. The novelty of this study should be inserted in the text clearly.

It is added in the abstract

  1. The abstract need to rewrite and the author/s should focus on the main findings and the main results.

Abstract has been rewritten and it is focused on those findings

  1. The lengthy sentences may be split into smaller sentence without change of its meaning.

14 The language of the whole manuscript is not very good and an improvement on the English language needs to be done in order to improve language of the manuscript.

These two questions are answered because we have rewritten the article with the help of a native English teacher

  1. Also, suggested to include the  references in the introduction part. In the introduction can be completed with properties and application of nano-size ceramics. The discussion can be extended by using the following paper: DOI: 10.1016/j.jallcom.2020.156695

We have include several new references. Our main objective is related with ceramic digital printing and it is referenced in number 23

  1. At the end of the introduction, the authors described the purpose (objectives) of the study, mentioning also the methods used, obviously briefly, but this should be explained in more detail in the methods.

Now, they are explained in the method part

  1. The methodology section is not well organized for the readers to understand the concept.

This section has been rewritten and with new organization step by stpe

  1. The author/s should be try to compare the results of the current study with the results of other researchers and it is required to explain if there is any agreement or disagreement.The results obtained should be compared with the in the literature. More care should be taken to present the results and facilitate understanding of the work

In this new version, a discussion comparing results with references has been stablished

  1. Figure 1 (XRD) has very poor resolution and is very loaded, to simplify and improve the resolution.

We have change resolution of figure 4 (first DRX) and 6 (second DRX) also we have changed axis without numeric scale.

  1. Compare XRD results with other articles.
  2. The XRD parameters compared with other literature studies.  Present the results obtained but do not compare with other existing studies. Are the values higher or lower than the literature? Add a comparative table.

We have added table 4 to make these comparation.

  1. Why is the size of the particles larger than the size of the crystallites?

Because we have great agglomeration due to high dwelling time during firing process

  1. In the conclusions appear aspects that should appear in the chapter of results and be described only strictly the conclusions of this study. In case of conclusions, you could explain what does it add to the subject area compared to other published material?

We have rewritten conclusions in order to include your suggestion.

  1. References are not written according to the guide (not all authors are listed in all references, they are not marked with initials, the titles of the articles and journals are not mentioned in many places, the volumes or pages are missing).

We are very sorry. We have added new references and according to the guide including DOI

Round 3

Reviewer 1 Report

The authors responded only partially to the observations made in the previous review.

I still believe that the reporting should be extended compared to the specialized literature and the degree of novelty of the article should be highlighted

Moderate editing of English language required

Author Response

Thank you very much for your review and for your effort doing it.

I am very sorry that your impression is that we have not answered all the questions, but we did answer them to the best of our ability.

In this review, the novelty of the article has been highlighted in the last paragraphs of the introduction, while we have included the bibliography that has been recommended to us in the synthesis and sample preparation section.

We trust that it will meet your expectations and if you have any other comments, we will be pleased to try to answer them.
